# Syntheses of l-Rhamnose-Linked Amino Glycerolipids and Their Cytotoxic Activities against Human Cancer Cells

**DOI:** 10.3390/molecules25030566

**Published:** 2020-01-28

**Authors:** Makanjuola Ogunsina, Pranati Samadder, Temilolu Idowu, Mark Nachtigal, Frank Schweizer, Gilbert Arthur

**Affiliations:** 1Department of Chemistry and Biochemistry, Faculty of Science, University of Manitoba, Winnipeg, MB R3T 2N2, Canada; mogunsina@gmail.com (M.O.); idowut@myumanitoba.ca (T.I.); 2Department of Biochemistry and Medical Genetics, Rady Faculty of Health Sciences, University of Manitoba, Winnipeg, MB R3E 0W9, Canada; sohagdalkutir@gmail.com (P.S.); Mark.Nachtigal@umanitoba.ca (M.N.); 3Department of Obstetrics, Gynecology and Reproductive Sciences, Rady Faculty of Health Sciences, University of Manitoba, Winnipeg, MB R3E 0W9, Canada; 4Research Institute in Oncology & Hematology, CancerCare Manitoba, Winnipeg, MB R3E 0V9, Canada

**Keywords:** glycosylated antitumor ether lipids, l-rhamnose-based glycolipids, chemotherapy resistant, caspase independent

## Abstract

A major impediment to successful cancer treatment is the inability of clinically available drugs to kill drug-resistant cancer cells. We recently identified metabolically stable l-glucosamine-based glycosylated antitumor ether lipids (GAELs) that were cytotoxic to chemotherapy-resistant cancer cells. In the absence of commercially available l-glucosamine, many steps were needed to synthesize the compound and the overall yield was poor. To overcome this limitation, a facile synthetic procedure using commercially available l-sugars including l-rhamnose and l-glucose were developed and the l-GAELs tested for anticancer activity. The most potent analog synthesized, 3-amino-1-*O*-hexadecyloxy-2R-(*O–*α-l-rhamnopyranosyl)-*sn*- glycerol **3**, demonstrated a potent antitumor effect against human cancer cell lines derived from breast, prostate, and pancreas. The activity observed was superior to that observed with clinical anticancer agents including cisplatin and chlorambucil. Moreover, like other GAELs, **3** induced cell death by a non-membranolytic caspase-independent pathway.

## 1. Introduction

Despite increasing investment in cancer research and treatment, the number of new cases and cancer deaths are increasing in both the developed and the developing world, affecting both males and females in nearly the same proportion [1]. Glycosylated antitumor ether lipids (GAELs), a subclass of antitumor ether lipids (AELs) [2,3], are cytotoxic to a wide range of epithelial cancer cell lines grown as adherent and spheroidal cultures as well as cancer stem cells (CSCs) [4,5,6,7,8,9,10]. While most anticancer drugs kill cells by inducing apoptosis, GAELs kill cells by an apoptosis independent mechanism [2,8,9], possibly by methuosis [11].

Recently, we reported the synthesis of GAELs with l-glucosamine as the sugar moiety [7]. We showed that in spite of the replacement of d-glucosamine in GAELs with the l-enantiomer, the compounds formed were equally cytotoxic to cancer cells [7]. The rationale for synthesizing the l-glucose analogues of the d-GAELs was to develop compounds that would be resistant to metabolic degradation (especially by glycosidases) as mammals do not have enzymes that metabolise l-glycosidic bonds. In the absence of commercially available l-glucosamine, synthesis of the l-glucosamine derived GAELs was initiated with l-mannose and required many synthetic steps. By contrast, l-sugars such as l-rhamnose or l-mannose that are commercially available are relatively simple to attach to a glycerol moiety. Our previous studies have shown that the activity of GAELs depends on the maintenance of a cationic amino group in the molecule [3]. The present study was undertaken to determine whether coupling the l-sugars to the glycerol moiety and maintaining the cationic amino moiety on the glycerol would generate biologically active l-GAELs with activity similar to the previously described l-glucosamine GAEL [7]. The advantage would be facile, cost effective synthetic procedures for potential anticancer compounds. Herein we report the synthesis of a l-rhamnose-GAEL compound and demonstrate its activity against human epithelial cancer cell lines.

## 2. Results

We have recently demonstrated that compound **1**, l-glucosamine GAEL (l-Gln) was cytotoxic against human epithelial cancer cells [7]. GAELs with l-sugars other than l-glucosamine (**1**) were synthesized to assess whether we could produce biologically active GAELs and generate insight on structural features required for activity. Novel l-GAEL analogues, compounds **2**–**6** (Figure 1), were synthesized and assessed for their cytotoxic activity. The neutral glycoglycerolipid **2** was synthesized to evaluate the effect of replacing l-glucosamine of compound **1** and d-glucosamine of **7** with the neutral sugar l-rhamnose. Compound **3** was synthesized to evaluate the effect of introducing a cationic moiety into l-rhamnose-based glycoglycerolipid. The primary position of the amine was selected for ease of synthesis. Compound **4** was synthesized to evaluate the effect of amide linkage with *p*-hydroxyphenyl propionic acid. The l-glucose-based cationic compound **5** was synthesized to evaluate the effect of different types of sugars relative to **3**. Compound **6** was synthesized to investigate the effect of the two amino moieties as prior studies showed that additional amino substituent at C-6 position of d-glucosamine-derived GAELs enhanced potency.

### 2.1. Synthesis of l-Sugar Derived Glycolipids 2–6

The hydroxyl groups of l-rhamnose **8** was protected with acetyl groups using acetic anhydride in pyridine and dimethylaminopyridine as catalyst to give the tetraacetate **9** (Scheme 1). The phenyl thioglycoside **10** was synthesized from **9** by a BF_3_·OEt_2_ promoted glycosylation with thiophenol. Silver triflate promoted glycosylation of **10** to the commercially available glycerolipid alcohol **11** yielded the fully protected glycolipid **12**. Sodium methoxide-catalysed deblocking of **12** in methanol gave the desired glycolipid **2** (Scheme 1).

Synthesis of compound **3** was achieved by glycosylation of the glycoside donor **10** with the acceptor azido lipid **15**, as described above, to give the fully protected azidoglycolipid **16**. The azidoglycerolipid **15** was made from the commercially available glycerol-analogue **13** as previously described [6]. Compound **16** was deacetylated to give the azide **17.** Reduction of the azido substituent of **17** was accomplished by using trimethyl phosphine in THF/water mixture to give monoamino compound **3** (Scheme 1). The amide **4** was synthesized by coupling **3** to a pre-activated *p*-hydroxyphenylproprionic acid **18** using TBTU [6] (Scheme 1).

The amino l-glucose based compound **5** was synthesized from l-glucose following the same procedure described above for the synthesis of compound **3** (Scheme 2).

Compound **6** was synthesized by coupling of glycoside donor **28** to lipid alcohol **15** to afford α-glycolipid **29** (Scheme 3). The glycoside donor **28** was synthesized from l-mannose in six steps. At first, the hydroxyl functional groups of l-mannose were acylated to afford pentaacetate **23** as described above. Compound **23** was subsequently converted into thiophenyl glycoside **24** by BF_3_·Et_2_O promoted glycosylation with thiophenol. Deacetylation of **24** gave compound **25**. The azido functional group at the C-6-position was installed by selective activation of the C-6 hydroxyl group in **25** as sulphonate ester to give **26**, followed by nucleophilic displacement of the sulphonate group by sodium azide in DMF to afford the 6-azido analog **27**. Protection of the remaining hydroxyl groups using acetic anhydride in pyridine produced the glycoside donor **28**. Donor **28** was used in glycosylation reaction with azido lipid alcohol **15** to produce glycolipid **29**. The ester groups were then deprotected to afford the diazido compound **30** which was subsequently subjected to reduced using trimethyl-phosphine in THF/water to produce the desired α-anomeric glycolipid **6** (Scheme 3).

### 2.2. In Vitro Screening of l-Sugar Derived Glycolipids’ Cytotoxic Activity Against Human Epithelial Cancer Cell Lines Derived from Breast, Prostate and Pancreas

The cytotoxic properties of compounds **2**–**6** were initially determined against exponentially growing human epithelial cancer cell lines including breast (MDA-MB-231, JIMT-1), pancreas (MiaPaCa2) and prostate (DU-145, PC-3) to identify the most active compound. The cells were incubated with increasing concentrations of the compounds (0–30 µM) for 48 h followed by assessment of cell viability using the MTS assay. The most potent of the newly synthesized compounds was the l-rhamno configured glycoglycerolipid **3**. This compound carries a primary amino group at the *sn*-3 position of the glycerolipid and l-rhamno sugar at the *sn*-2 position via an α-rhamnosidic linkage. Compound **3** was active with CC_50_ values of 4.8–14 μM (Figure 2 and Table 1). Greater than 90% loss of cell viability was observed at a concentration range of 7.5–15.0 µM for all the cell lines. Out of these five cell lines, MDA-MB-231, a cisplatin resistant and triple negative breast cancer cell line, was most sensitive to compound **3**.

The second most potent compound was the l-manno configured bisamine **6** which in contrast to **3** bears a primary amino substituent at the position C-6 of the sugar. This compound was typically 2- to 4-times less active than glycolipid **3** with CC_50_ values in the range of 12.5–25 µM (Table 1). This is in contrast to previous observation with l-glucosamine derived GAELs where a primary amino substituent at the C-6 of the l-glucose scaffold enhanced activity [7]. But it is noteworthy that in the d-sugar GAEL series, there was significant loss of activity with mannose analogs [10], CC_50_ was not reached at 30 µM.

The l-rhamnose based analog **2** that is devoid of any amino group and amide **4** were significantly less active than **3** and **6** with CC_50_ values ≥ 21.5 µM across all the five cell lines. The reduced cytotoxicity of compounds **2** and **4** when compared to that of compound **3** demonstrated that the free amine is an important structural feature required for cytotoxicity of l-sugar-derived GAELs. This is consistent with earlier results on d-glucosamine-derived GAEL series [10]. Surprisingly, compound **5**, an analog of **3** where the l-rhamnose is replaced with l-glucose is was less active. Depending on the cell line, loss of viability caused by **5** ranged from 0–35%. We were therefore unable to achieve CC_50_ at the highest concentration tested (30 µM) for any of the cell lines.

The effects of l-mannose, l-glucose, rhamnose (**8**) were also tested to determine if these l-sugars displayed any cytotoxicity against the cancer cells. The results revealed that rhamnose, l-mannose and l-glucose were not cytotoxic at the highest dose (30 µM) tested (data not shown).

The activity of the most active analogue, l-rhamnose linked glycerolipid **3,** was compared with results published on the activity of l- and d-glucosamine based analogs **1** and **7 [7]**. Compound **3** is as active or more active (CC_50_ = 4.8–11.0 µM) than **1** (6.5–12 µM) or **7** (8.0–13.5 µM) across all the cell lines (Table 1).

The activities of compounds **2**–**6** against the cancer cell lines were compared with those of clinically-used anticancer agents: chlorambucil and cisplatin. As shown in Table 1, compounds **2**–**6** were more potent across the five cell lines than chlorambucil. A CC**_50_** concentration for chlorambucil was not attained even at a concentration as high as 150 µM. Compounds **3** and **6** also displayed superior cytotoxicity than cisplatin against all the cell lines. Cisplatin did not achieve 50% cell death at 20 µM against MDA-MB-231 and MiaPaCa-2 cell lines. Thus, the activity of **3** in vitro was better than that of anticancer drugs like chlorambucil and cisplatin. The activity of **3** was similar or better than that of l-glucosamine derived compound **1**.

Having identified **3** as the most active of the synthesised l-sugar analogues, its effects on the viability of additional cancer cell lines including BT549, MDA-MB-453, MDA-MB-468, and Hs578t (triple negative breast cancer lines), and BT474 (hormone receptor positive breast cancer cells) (Figure 3) was investigated. Concentrations of compound **3** between 6–15 µM were sufficient to kill all the triple negative cell lines with the exception of MDA-MB-453 where viability was inhibited by 60% at the highest concentration tested. 30 µM of **3** was required for complete loss of viability of the hormone sensitive breast cancer BT474 cell line.

### 2.3. Determination of the Effect of Pan-caspases Inhibition on Cytotoxicity of Compound 3 in JIMT-1 and DU-145 Cell Lines

One of the hallmark features of cytotoxic GAELs is their ability to kill cancer cells via an apoptosis-independent mechanism [2,4,5,7,8,9]. We evaluated whether the mechanism of action of **3** was similar to other GAELs. The ability of **3** to kill JIMT-1 and DU-145 cell lines in the presence or absence of QVD-OPh (QVD), an effective pan-caspase inhibitor [12], was investigated. There were no statistically significant differences in cytotoxicity in the absence or presence of 40 µM QVD (Figure 4). In contrast, QVD significantly reduced cytotoxicity of Adriamycin, an apoptosis inducing anticancer agent, in JIMT-1 and DU-145 cell lines at all concentration tested. The results of the studies show that compound **3** kills cancer cells via a caspase-independent pathway(s).

### 2.4. Evaluation of Effect of l-Rhamnose Based GAEL 3 on Integrity of DU145 or JIMT-1 Cell Membranes

After determining that the mechanism of cell death of compound **3** is independent of caspases-mediated apoptosis, we evaluated if cell death was due to membrane disruption or lysis using the cell impermeant ethidium homodimer-1 (EthD-1) dye that emits red fluorescence upon binding to DNA [13,14]. DU-145 and JIMT-1 were treated with compound **3** at 10 and 15 µM, respectively. After 4–6 h, the cells were treated with EthD-1 dye and monitored under fluorescence microscopy. Comparison of untreated control cells and cells treated with compound **3** in both DU-145 and JIMT-1 cell lines demonstrated that there was no significant cell membrane lysis. In contrast, cells incubated with 0.01% Triton X-100 for 10 min stained bright red (Figure 5).

### 2.5. Hemolytic Properties of l-Rhamno Configured GAEL 3

To further evaluate the membrane lytic properties of compound **3**, we investigated its capacity to lyse ovine erythrocytes [13]. The highest hemolysis observed with compound **3** was about 6.6% at 100 µM compared to the 1% ammonium hydroxide positive control (Figure 6). These results indicated that compound **3** is not lytic and may be compatible with intravenous administration without concern for hemolysis.

## 3. Discussion

GAELs are a promising class of investigational anticancer agents with potent antitumor activity against a range of cancer cell lines and CSCs [2,3,5,6]. The biologically active GAELs that we have developed thus far typically have a glycerol backbone with a methoxy group at the C-2 position, a d- or l-glucose at the C-3 position, and an amino group attached to the sugar [3,4,6,7]. Due to the relative expense and complexity in synthesizing l-glucosamine GAEL analogues, the objective of the present study was to develop biologically active GAELs with l-sugars via a facile approach that would facilitate large scale synthesis of compounds for in vivo studies. Our previous studies identified that the presence of a primary amine in the molecule is essential for good cytotoxic activity. To simplify synthesis, the amino group was placed at the C-3 position of the glycerol backbone while the l-sugar was placed at the C-2 position. It was reasoned that this would satisfy the requirement for a free amine needed for activity. This was tested with the synthesis of l-rhamnose GAEL (**3**) and l-glucose-GAEL (**5**). The results showed that this GAEL scaffold did in fact result in bioactive compounds. Comparison of the activity of the novel compounds synthesised (**2**–**6**) against epithelial cancer cell lines derived from breast, pancreas and prostate tumors, revealed that the most potent analog was the l-rhamnose configured analog **3**. The l-rhamnose based analog **2**, which does not have a primary amine, or compound **4** with an amide linkage, were significantly less potent than **3** with a primary amino substituent and l-rhamnose at position *sn*-3 and *sn*-2 of the glycerolipid, respectively. This observation shows that the primary amino substituent is essential for cytotoxicity of this series of GAELs and is akin to what was observed with glucose-based GAELs which required an amino group on the sugar [2,6,10]. However, compound **5**, an analog in which the l-rhamnose of **3** was replaced with l-glucose showed minimal cytotoxicity. Indeed, CC_50_ values were not attained at the highest tested concentration for all the cell lines. Thus, the nature and type of sugar moiety in GAELs play a crucial role in their cytotoxicity. We have previously demonstrated with the GAEL series that the nature of the sugar also affects activity [10].

Compound **3** was active against a wide range of breast cancer cells differing in hormone receptor status suggesting that the presence or absence of the receptors was not a determinant in the susceptibility of the cells to compound **3**. Thus, BT-474 hormone receptor (HER/2, ER, PR) positive cells and the MDA-MB-453, a TNBC cell line, had similar CC_50_’s of 14 µM vs. 13.8 µM. Also, JIMT-1 cell line (overexpression of Her2) was more sensitive to compound **3** than some TNBC cell lines.

Studies conducted to provide insight on the mode of cell death induced by **3** revealed that death occurs via an apoptosis-independent pathway. Thus, the mechanism of action of **3** may be similar to that described for glucose-based GAELs [4,7,8,9,11]. This non-apoptotic mechanism of cell death may explain why GAELs are effective against resistant cancer lines as cells escape cell death through many mechanisms including over expression of proteins that inhibit caspases and apoptosis [15,16,17]. Also, the result of membrane disruption experiment using cell impermeant EthD-1 dye demonstrated that **3** does not kill cells by membrane disruption. The fact that the cells were rounding up as they were dying may imply possible interference with the cell cytoskeleton. Our hemolysis assay on ovine erythrocytes further confirmed that l-sugar GAELs induced cell death is not due to interference with cell membrane integrity. The results also indicate that l-rhamnose GAEL **3** may be safely delivered by intravenous administration.

In conclusion, this study identified a novel scaffold for active l-GAELs. The most active compound was l-rhamnose GAEL **3** that has potent cytotoxicity against a spectrum of cancer cell lines. The ease of synthesis makes **3** a suitable compound for large scale synthesis for preclinical studies in rodent models of human cancer.

## 4. Materials and Methods

### 4.1. Synthesis of GAELs

Solvents were dried over CaH_2_. ^1^H-, ^13^C-NMR spectra were recorded on an Avance 300 NMR spectrometer (Bruker, Rheinstetten, Germany), and chemical shifts reported (in ppm) relative to internal Me_4_Si (δ = 0.0) and at 75 MHz, respectively. NMR spectra (^1^H-NMR, ^13^C-NMR) of compounds **2**–**6** are provided in the Appendix A. Thin-layer chromatography (TLC) was carried out on aluminum or glass-backed silica gel GF plates (250 µm thickness) and plates were visualized by charring with 5% H_2_SO_4_ in MeOH and or short wavelength UV light. Compounds were purified by flash chromatography on silica gel 60 (230–400 ASTM mesh). ESI-MS analyses were performed on a 500 MS Ion Trap Mass Spectrometer (Varian, Walnut Creek, CA, USA). MALDI-TOF-MS were performed on a Bruker Daltonics Ultraflex MALDI TOF/TOF Mass Spectrometer (Bremen, Germany). Purity of compound **2**–**6** was assessed by ^1^H-NMR spectroscopy after reverse phase chromatography.

1,2,3,4-l-Rhamnopyranosyl Tetraacetate (**9**)

l-Rhamnose **8** (0.99 g, 5.49 mmol) was dissolved in 20.0 mL pyridine at room temperature, then acetic anhydride (5.20 mL, 54.90 mmol) and dimethyl amino pyridine (0.10 g) were sequentially added and the reaction was vigorously stirred for 18 h after which it was stopped by addition of methanol (10.0 mL) and then stirred for 15 min. The solvents were removed under high vac. The resulting residue was then dissolved in ethyl acetate (50.0 mL) and washed with 3% HCl solution (1 time), saturated sodium bicarbonate (2 times), distilled water (1 time) and brine (1 time). The resulting organic layer was dried over Na_2_SO_4_ and concentrated to dryness and purified by flash chromatography using ethyl acetate and hexane (1:1) to give **9** (1.71 g, 5.12 mmol) as α, β mixture (9:1). Yield was 93%. NMR data for α- anomer of compound **9**: ^1^H-NMR (300 MHz, chloroform-*d*) δ 6.01 (d, *J* = 1.8 Hz, 1H, H-1), 5.31 (dd, *J* = 10.1, 3.5 Hz, 1H, H-3), 5.25 (dd, *J* = 3.5, 1.8 Hz, 1H, H-2), 5.12 (dd, *J* = 10.1, 9.9 Hz, 1H, H-4), 3.93 (m, 1H), 2.17 (s, 3H), 2.16 (s, 3H), 2.06 (s, 3H), 2.00 (s, 3H), 1.24 (d, *J* = 6.2 Hz, 3H, H-6). ^13^C-NMR (75 MHz, CDCl_3_) δ 170.05, 169.81, 169.79, 168.35, 90.65, 70.48, 68.77, 68.72, 68.65, 20.89, 20.77, 20.74, 20.67, 17.44. ES-MS: calculated (calcd): C_14_H_20_O_9_Na^+^
*m*/*z*: 355.1, found [M + Na]^+^
*m*/*z*: 355.5.

Phenyl-2,3,4-triacetyl-1-thio-α-l-rhamnopyranoside (**10**)

The tetraacetate **9** (1.71 g, 5.12 mmol) was dissolved in in 20.0 mL DCM, then thiophenol (0.68 g, 6.14 mmol) and BF_3_·Et_2_O (0.87 g, 6.14 mmol) were sequentially added. The reaction was stirred vigorously for 18 h after which it was stopped with 20 mL saturated sodium bicarbonate at 0 °C. The organic layer was separated using separatory funnel and subsequently washed with saturated 20.0 mL of sodium bicarbonate (2 times), 20.0 mL of water (1 time) and 20.0 mL of brine (1 time). The organic layer was dried over anhydrous sodium bicarbonate and then concentrated under vacuum. The residue was then purified by flash chromatography using ethyl acetate and hexane (4:6) to give mainly α-anomer of **10** (1.72 g, 4.5 mmol). Yield was 88%. NMR data for the anomer of compound **10**: ^1^H-NMR (300 MHz, chloroform-*d*) δ 7.59–7.22 (m, 5H, aromatic protons), 5.51 (dd, *J* = 3.4, 1.1 Hz, 1H, H-2), 5.42 (d, *J* = 1.1 Hz, 1H, H-1), 5.30 (dd, *J* = 10.0, 3.4 Hz 1H, H-3), 5.19 (dd, *J* = 9.6, 10.0 Hz, 1H, H-4), 4.42–4.34 (m, 1H, H-5), 2.15 (s, 3H), 2.09 (s, 3H), 2.05 (s, 3H), 1.26 (d, *J* = 6.2 Hz, 3H, H-6). ^13^C- NMR (75 MHz, CDCl_3_) δ 169.99, 169.98, 169.91, 132.08, 131.85, 129.19, 127.89, 85.71, 71.34, 71.17, 69.40, 67.79, 20.91, 20.82, 20.69, 17.35. ES-MS: calcd: C_18_H_22_O_7_SNa^+^*m*/*z*: 405.1, found [M + Na]^+^
*m*/*z*: 405.3.

1-Hexadecyloxyl-2*R*-methoxyl-3-(2′3′4′-triacetyl-α-l-rhamnopyranosyl)-*sn*-glycerol (**12**)

The fully protected glycoside donor **10** (153 mg, 0.4 mmol) and the glycoside acceptor **11** (140 mg, 0.4 mmol) were dissolved in 15 mL of DCM under argon atmosphere, then AgOTf (0.02 g, 0.08 mmol) and *N*-iodosuccinimide (0.14 g, 0.60 mmol) were simultaneously added. The reaction was vigorously stirred for 2 h after which it was stopped with saturated solution of sodium thiosulphate (5.0 mL) and then washed with 25.0 mL of saturated sodium thiosulphate solution (1 time), saturated sodium bicarbonate (3 times), water (1 time) and brine (1 time). The organic layer was then dried over anhydrous sodium sulphate and then concentrated under vac. The residue, was purified by flash chromatography using ethyl acetate/hexane mixture (4:6) to give the α-anomer, **12** (0.19 g, 0.31 mmol) as a white solid. Yield was 58%. NMR data of **12**: ^1^H-NMR (300 MHz, chloroform-*d*) δ 5.36–5.19 (m, 2H, H-2, H-3), 5.04 (dd, *J* = 9.7, 9.7 Hz, 1H, H-4), 4.73 (d, *J* = 1.5 Hz, 1H, H-1), 3.93–3.84 (m, 1H, H-5), 3.79–3.63 (m, 1H), 3.61–3.34 (m, 9H), 2.12 (s, 3H), 2.02 (s, 3H), 1.96 (s, 3H), 1.50–1.48 (m, 2H), 1.23 (broad s, 26H, lipid tail), 1.20 (d, *J* = 6.2 Hz, 3H, H-6), 0.85 (t, *J* = 6.6 Hz, 3H, terminal lipid CH_3_). ^13^C-NMR (75 MHz, CDCl_3_) δ 170.01, 169.92, 169.87, 97.58, 78.94, 71.78, 71.12, 69.80, 69.67, 69.12, 67.35, 66.34, 58.20, 31.89, 29.66, 29.62, 29.47, 29.33, 26.09, 22.65, 20.85, 20.74, 20.67, 17.39, 14.08. ES-MS: calcd: C_32_H_58_O_10_Na^+^
*m*/*z*: 625.4, found [M + Na]^+^
*m*/*z*: 624.8.

1-Hexadecyl-2*R*-methoxyl-3-*O*-α-l-rhamnopyranosyl-*sn*-glycerol (**2**)

Compound **12** (0.19 g, 0.31 mmol) was dissolved in 15.0 mL of methanol, then catalytic amount of sodium methoxide was added and the reaction was vigorously stirred for 3 h. The reaction was stopped by acidic ion exchange resin. The resin was filtered and the filtrate was concentrated under vacuum and the residue was purified by flash chromatography using ethyl acetate/hexane mixture (9:1) to give **2** (0.11 g, 0.23 mmol) as a white solid. Yield was 75%. NMR data of **2**: ^1^H-NMR (300 MHz, chloroform-*d*) δ 4.78 (d, *J* = 1.1, 1H, H-1), 4.12 (s, 3H, OH, rhamnose-OH), 3.97 (dd, *J* = 1.1, 3.3 Hz, 1H, H-2), 3.83–3.63 (m, 3H), 3.63–3.53 (m, 1H), 3.53–3.36 (m, 9H), 1.56 (m, 2H), 1.31 (d, *J* = 6.0 Hz, 3H, H-6), 1.28 (broad s, 26H, lipid tail), 0.89 (t, *J* = 6.5 Hz, 3H, terminal lipid CH_3_). ^13^C-NMR (75 MHz, CDCl_3_) δ 99.92, 79.03, 72.80, 71.84, 71.68, 70.89, 69.99, 68.24, 66.71, 58.04, 31.94, 29.72, 29.69, 29.37, 26.13, 22.70, 17.55, 14.12. MALDI-HRMS: calcd: C_26_H_52_O_7_Na^+^
*m*/*z*: 499.3611, found [M + Na]^+^
*m*/*z*: 499.3615.

3-Hexadecyloxy-2*R*-hydroxyl propyl-1-*p*-toluene sulphonate (**14**)

The lipid diol **13** (2.00 g, 6.32 mmol) was dissolved in dissolved in 20 mL DCM, cooled to 0 °C, then Et_3_N (1.28 g, 1.80 mL,) was added followed by 4-toluenesulphonyl chloride (1.33 g, 6.95 mmol) and DMAP (0.04 g, 0.32 mmol). The temperature was allowed to increase to room temperature (23 °C) and the mixture was stirred for 4 h. At the end of reaction, the mixture was diluted with ethyl (60.0 mL) acetate, washed with saturated aqueous ammonium chloride (3 times), brine (3 times). The organic layer was then dried over sodium sulphate, concentrated under vacuum and the residue was purified using flash chromatography using hexane/ethyl acetate (8:2) to give **14** (1.80 g, 3.80 mmol) as a white flaky solid. Yield was 60%. NMR data of **14**: ^1^H NMR (300 MHz, chloroform-*d*) δ 7.78 (d, *J* = 8.2 Hz, 2H, aromatic protons), 7.33 (d, *J* = 8.1 Hz, 2H, aromatic protons), 4.11–4.00 (m, 2H, TsO-CH_2_), 3.99–3.89 (m, 1H, HO-CH), 3.46–3.31 (m, 4H), 2.80 (d, *J* = 5.4 Hz, 1H, OH), 2.42 (s, 3H, toluene-CH_3_), 1.55–1.41 (m, 2H), 1.25 (s, 26H, Lipid tail), 0.87 (t, *J* = 6.4 Hz, 3H, lipid terminal-CH_3_).^13^C-NMR (75 MHz, CDCl_3_) δ 144.90, 132.77, 129.88, 127.99, 71.73, 70.77, 70.56, 68.25, 31.93, 29.71, 29.68, 29.64, 29.61, 29.48, 29.37, 26.01, 22.68, 21.58, 14.11. ES-MS: calcd: C_26_H_46_NO_5_Na^+^
*m*/*z*: 493.3, found [M + Na]^+^
*m*/*z*: 493.7.

3-Hexadecyloxy-2*R*-hdroxyl propyl-1-azide (**15**)

Compound **14** (1.30 g, 2.76 mmol) and sodium azide (1.81 g, 27.60 mmol) were suspended in anhydrous DMF and the mixture was stirred at 90 °C for 18 h. at the end of the reaction the mixture was concentrated then diluted with ethyl acetate and filtered to remove excess sodium azide. The filtrate was then concentrated and purified with flash chromatography using hexane/ethyl acetate (9:1) to give **15** (0.85 g, 2.50 mmol) as a white wax like solid. Yield was 91%. ^1^H-NMR (300 MHz, Chloroform-*d*) δ 3.92–3.86 (m, 1H, HO-CH), 3.48–3.34 (m, 4H), 3.31 (dd, *J* = 5.5, 2.9 Hz, 2H, -CH_2_N_3_), 3.17 (s, 1H, OH), 1.55–1.41 (m, 2H, 1.25 (s, 26H, Lipid tail)), 0.85 (t, *J* = 6.6 Hz, 3H, terminal lipid-CH_3_).^13^C-NMR (75 MHz, CDCl_3_) δ 71.92, 71.71, 69.59, 53.54, 31.93, 29.71, 29.67, 29.61, 29.52, 29.47, 29.37, 26.05, 22.67, 14.03.ES-MS: calcd: C_19_H_39_N_3_O_2_Na^+^
*m*/*z*: 364.3, found [M + Na]^+^
*m*/*z*: 364.5.

3-Azido-1-hexadecyloxyl-2*R*-(2′3′4′-tri-*O*-acetyl-*O*-α-l-rhamnopyranosyl)-*sn*-glycerol (**16**)

The fully protected glycoside donor **10** (0.15 g, 0.41 mmol) and the glycoside acceptor **15** (0.12 g, 0.36 mmol) were dissolved in 15.0 mL of DCM under argon atmosphere, then AgOTf (0.02 g, 0.08 mmol) and *N*-iodosuccinimide (0.14 g, 0.60 mmol) were simultaneously added. The reaction was vigorously stirred for 2 h after which it was stopped with saturated solution of sodium thiosulphate (5.0 mL) and then washed with 25.0 mL of saturated sodium thiosulphate solution (1 time), saturated sodium bicarbonate (3 times), water (1 time) and brine (1 time). The organic layer was then dried over anhydrous sodium sulphate and then concentrated under vac. The residue was purified by flash chromatography using ethyl acetate/hexane mixture (4:6) to give α-anomers, **16** (0.12 g, 0.20 mmol) as a white solid. Yield was 55%. NMR data of **16:**
^1^H-NMR (300 MHz, chloroform-*d*) δ 5.30 (dd, *J* = 10.0, 3.6 Hz, 1H, H-3), 5.25 (dd, *J* = 3.6, 1.7 Hz, 1H, H-2), 5.06 (dd, *J* = 9.8, 9.9 Hz, 1H, H-4), 4.93 (d, *J* = 1.7 Hz, 1H, H-1), 4.18–3.99 (m, 1H, H-5), 3.95–3.83 (m, 1H), 3.58–3.29 (m, 6H), 2.14 (s, 3H), 2.03 (s, 3H), 1.98 (s, 3H), 1.57–1.52 (m, 2H), 1.25 (broad s, 26H, lipid tail), 1.20 (d, *J* = 6.3 Hz, 3H, H-6), 0.87 (t, *J* = 6.6 Hz, 3H). ^13^C-NMR (75 MHz, CDCl_3_) δ 170.01, 169.95, 169.84, 97.22, 76.46, 71.77, 71.09, 70.48, 70.01, 68.92, 66.68, 51.68, 31.91, 29.68, 29.49, 29.34, 26.13, 20.87, 20.75, 20.67, 17.34, 14.09. ES-MS: calcd: C_31_H_55_N_3_O_9_Na^+^
*m*/*z*: 636.4, found [M + Na]^+^
*m*/*z*: 636.5.

3-Azido-1-hexadecyloxyl-2*R*-*O*-α-l-rhamnopyranosyl-*sn*-glycerol (**17**)

Compound **16** (0.12 g, 0.20 mmol) was dissolved in 15.0 mL of methanol, then catalytic amount of sodium methoxide was added and the reaction was vigorously stirred for 3 h. The reaction was stopped by acidic ion exchange resin. The resin was filtered and the filtrate was concentrated under vacuum and the residue was purified by flash chromatography using ethyl acetate/hexane mixture (9:1) to give **17** (0.08 g, 0.16 mmol) as a white solid. Yield was 79%. NMR data of **17**: ^1^H-NMR (300 MHz, chloroform-*d*) δ 4.95 (d, *J* = 1.1, 1H, H-1), 4.19–3.95 (m, 1H, H-5), 4.03–3.85 (m, 2H), 3.77 (d, *J* = 8.3, 3.5 Hz, 1H, H-3), 3.62–3.27 (m, 10H), 1.58–1.54 (m, 2H), 1.32 (d, *J* = 6.4 Hz, 3H, H-6), 1.27 (broad s, 26H), 0.88 (d, *J* = 7.1 Hz, 3H). ^13^C-NMR (75 MHz, CDCl_3_) δ 100.04, 76.26, 72.70, 71.83, 71.60, 71.09, 70.33, 68.67, 51.71, 31.94, 29.73, 29.52, 29.38, 26.11, 22.70, 17.48, 14.12. ES-MS: calcd: C_25_H_49_N_3_O_6_Na^+^
*m*/*z*: 500.4, found [M + Na]^+^
*m*/*z*: 500.4.

3-Amino-1-*O*-hexadecyloxy-2*R*-(*O*–α-l-rhamnopyranosyl)-*sn*-glycerol (**3**)

To a solution compound **17** (0.10 g, 0.21 mmol) in THF (7.0 mL) was added 1.5 mL of water and 2.6 mL of 1 M trimethylphosphine in THF. The reaction was vigorously stirred for 2 h at room temperature after which it was concentrated under vac. The residue was purified by C-18 column using gradient elution with water/methanol to give **3** (0.06 g, 0.13 mmol) as a white solid. Yield was 61%. NMR data for **3**: ^1^H-NMR (300 MHz, methanol-*d*_4_) δ 4.65 (d, *J* = 1.3 1H, H-1) 3.65 (dd, *J* = 1.3, 3.4 Hz, 1H, H-2), 3.48–3.56 (m, 2H), 3.45 (dd, *J* = 9.5, 3.4 Hz, 1H, H-3), 3.37–3.29 (m, 1H, H-5), 3.29–3.11 (m, 4H), 2.59–2.42 (m, 2H), 1.40–1.34 (m, 2H), 1.08 (broad s, 29H, H-6, lipid tail), 0.69 (t, *J* = 6.4 Hz, 3H, lipid terminal-CH_3_). ^13^C-NMR (75 MHz, MeOD) δ 101.91, 79.55, 73.98, 72.69, 72.66, 72.45, 72.39, 70.10, 43.50, 33.10, 30.81, 30.78, 30.66, 30.50, 27.33, 23.76, 18.08, 14.47. MALDI-HRMS: calcd: C_25_H_51_NO_6_Na^+^
*m*/*z*: 484.3614, found [M + Na]^+^
*m*/*z*: 484.3611.

3-(-3-(*p*-Hydroxyphenylpropyl)-amido-1-*O*-Hexadecyloxy-2*R*-(*O*-α-l-rhamnopyranosyl)-*sn*-glycerol (**4**)

To a solution of p-hydroxyphenyl propionic acid **18** (0.02 g, 0.11 mmol), TBTU (0.05 g, 0.14 mmol) and diisopropyl ethyl amine (0.02 g, 0.14 mmol) in 5.0 mL of DMF which has been stirring for 20 min was added **3** (0.05 g, 0.11 mmol). The reaction was vigorously stirred for 8 h after which it was diluted with methanol. The solvents were removed in vacuo and the residue was purified by flash chromatography using ethylacetate to give **4** (0.07 g, 0.11 mmmol) as an off white solid. Yield was 98%. NMR data of **4**: ^1^H-NMR (300 MHz, methanol-*d*_4_) δ 6.94 (d, *J* = 8.3 Hz, 2H, aromatic proton), 6.62 (d, *J* = 8.3 Hz, 2H, aromatic proton), 4.72 (d, *J* = 1.9 Hz, 1H, H-1), 3.80–3.62 (m, 2H), 3.55 (dd, *J* = 9.8, 5.7 Hz, 1H), 3.41–3.21 (m, 7H), 3.16 (dd, *J* = 13.8, 5.6 Hz, 1H), 2.73 (t, *J* = 7.5 Hz, 2H, propionamide CH_2_), 2.36 (t, *J* = 7.7 Hz, 2H, propionamide CH_2_), 1.49–1.41 (m, 2H), 1.21 (broad s, 26H, lipid tail), 1.16 (d, *J* = 6.2 Hz, 3H, H-6), 0.82 (t, *J* = 6.4 Hz, 3H, lipid terminal CH_3_). ^13^C-NMR (75 MHz, MeOD) δ 175.67, 156.96, 132.79, 130.30, 116.30, 101.26, 76.32, 74.02, 72.67, 72.53, 72.34, 70.00, 49.89, 48.18, 40.92, 39.32, 33.10, 32.20, 30.82, 30.70, 30.50, 27.32, 23.76, 18.09, 14.48. MALDI-HRMS: calcd: C_34_H_59_NO_7_Na^+^
*m*/*z*: 632.4138, found [M + Na]^+^
*m*/*z*: 632.4590.

α/β-l-Glucopyranosyl-1,2,3,4,5-pentaacetate (**19**)

l-Glucose (0.90 g, 5.00 mmol) was dissolved in 20.0 mL pyridine at room temperature, then acetic anhydride (5.20 mL, 54.90 mmol) and dimethyl amino pyridine (0.10 g,) were sequentially added and the reaction was vigorously stirred for 18 h after which it was stopped by addition of methanol (10.0 mL) and then stirred for 15 min. The solvents were removed under high vac. The resulting residue was then dissolved in ethyl acetate (50.0 mL) and washed with 3% HCl solution (1 time), saturated sodium bicarbonate (2 times), distilled water (1 time) and brine (1 time). The resulting organic layer was dried over Na_2_SO_4_ and concentrated to dryness and purified by flash chromatography using ethyl acetate and hexane (1:1) to give **19** (1.70 g, 4.40 mmol) as α, β mixture (3:2). Yield was 88%. Characteristic proton NMR data for **19**: ^1^H-NMR (300 MHz, chloroform-*d*) δ 6.35 (d, *J* = 3.7 Hz, 3H, α-H-1), 5.73 (d, *J* = 8.2 Hz, 2H, β-H-1). ES-MS: calcd: C_16_H_22_O_11_Na^+^
*m*/*z*: 413.1, found [M + Na]^+^
*m*/*z*: 413.4.

Phenyl-2,3,4,6-tetra-*O*-acetyl-1-thio-β-l-glucopyranoside (**20**)

The pentaaacetate **19** (1.70 g, 4.40 mmol) was dissolved in in 20.0 mL DCM, then thiophenol (0.68 g, 6.14 mmol) and BF_3_·Et_2_O (0.87 g, 6.14 mmol) were sequentially added. The reaction was stirred vigorously for 18 h after which it was stopped with 20.0 mL saturated sodium bicarbonate at 0 °C. The organic layer was separated using separatory funnel and subsequently washed with saturated 20.0 mL of sodium bicarbonate (2 times), 20.0 mL of water (1 time) and 20.0 mL of brine (1 time). The organic layer was dried over anhydrous sodium bicarbonate and then concentrated in vacuo. The residue was then partially purified by flash chromatography using ethyl acetate and hexane (4:6) to give mainly β-anomer of **20** (1.72 g, 4.50 mmol). Yield was 88%. Compound **20** was not characterised.

3-Azido-1-hexadecyloxyl-2*R*-(2′3′4′6′-tetra-*O*-acetyl-*O*-β-l-glucopyranosyl)-*sn*-glycerol (**21**)

The fully protected glycoside donor **20** (0.18 g, 0.40 mmol) and the glycoside acceptor **15** (0.15 g, 0.44 mmol) were dissolved in 15.0 mL of DCM under argon atmosphere, then AgOTf (0.02 g, 0.08 mmol) and *N*-iodosuccinimide (0.18 g, 0.80 mmol) were simultaneously added. The reaction was vigorously stirred for 2 h after which it was stopped with saturated solution of sodium thiosulphate (5.0 mL) and then washed with 25.0 mL of saturated sodium thiosulphate solution (1 time), saturated sodium bicarbonate (3 times), water (1 time) and brine (1 time). The organic layer was then dried over anhydrous sodium sulphate and then concentrated under vac. The residue was purified by flash chromatography using ethyl acetate/hexane mixture (4:6) to give **21** (0.13g, 0.20 mmol) as a white solid. Yield was 50%. ^1^H-NMR (300 MHz, chloroform-*d*) δ 5.22 (dd, *J* = 9.4, 9.4 Hz, 1H, H-4), 5.10 (dd, *J* = 9.6, 9.6 Hz, 1H, H-3), 4.97 (dd, *J* = 9.6, 7.9 Hz 1H, H-2), 4.75 (d, *J* = 7.9 Hz, 1H, H-1), 4.25–4.18 (m, 2H), 4.02–3.90 (m, 1H), 3.78–3.69 (m, 1H), 3.53–3.23 (m, 6H), 2.10 (s, 3H), 2.05 (s, 3H), 2.04 (s, 3H), 2.02 (s, 3H), 1.59–1.52 (m, 2H), 1.27 (broad singlet, 26H, Lipid tail), 0.89 (h, *J* = 6.1 Hz, 3H, Lipid terminal-CH_3_). ^13^C-NMR (75 MHz, CDCl_3_) δ 170.65, 170.29, 169.37, 169.17, 100.44, 77.64, 72.83, 71.93, 71.85, 71.47, 70.58, 68.47, 61.90, 52.06, 31.93, 29.70, 29.66, 29.61, 29.49, 29.36, 26.12, 22.70, 20.72, 20.62, 14.12. ES-MS: calcd: C_33_H_57_N_3_O_11_Na^+^
*m*/*z*: 694.4, found [M + Na]^+^
*m*/*z*: 694.8.

3-Azido-1-hexadecyloxyl-2*R*-*O*-β-l-glucopyranosyl-*sn*-glycerol (**22**)

Compound **21** (0.13 g, 0.20 mmol) was dissolved in 15.0 mL of methanol, then the catalytic amount of sodium methoxide was added and the reaction was vigorously stirred for 3 h. The reaction was stopped by acidic ion exchange resin. The resin was filtered and the filtrate was concentrated under vacuum and the residue was dissolved in ethyl acetate and filtered through a pad of silica gel to give **22** (0.09 g, 0.17 mmol) as a white solid. Yield was 84%. Compound **22** was not characterised. ES-MS: calcd: C_25_H_49_N_3_O_7_Na^+^
*m*/*z*: 526.4, found [M + Na]^+^
*m*/*z*: 526.5.

3-Amino-1-hexadecyloxyl-2*R*-*O*-β-l-glucopyranosyl-*sn*-glycerol (**5**)

To a solution of compound **22** (0.09 g, 0.17 mmol) in THF (7.0 mL) was added 1.5.0 mL of water and 2.6 mL of 1 M trimethylphosphine in THF. The reaction was vigorously stirred for 2 h at room temperature after which it was concentrated under vac. The residue was purified by C-18 column using gradient elution with water/methanol to give **5** (0.06 g, 0.09 mmol) as a white solid. Yield was 55%. NMR data for **5**: ^1^H-NMR (300 MHz, methanol-*d*_4_) δ 4.35 (d, *J* = 7.7, Hz, 1H, H-1), 3.81 (dd, *J* = 13.7, 3.6 Hz, 2H, -CH-CH_2_O-), 3.69–3.36 (m, 6H), 3.34–3.07 (m, 4H), 1.59–1.38 (m, 3H), 1.22 (broad s, 26H, Lipid tail), 0.85 (t, *J* = 7.3, 3H, Lipid terminal-CH_3_). ^13^C-NMR (75 MHz, MeOD) δ 103.87, 79.65, 78.17, 77.96, 75.13, 72.76, 72.35, 71.66, 62.80, 43.87, 33.09, 30.81, 30.78, 30.49, 27.26, 23.75, 14.46. MALDI-HRMS: calcd: C_25_H_51_NO_7_Na^+^
*m*/*z*: 500.3563, found [M + Na]^+^
*m*/*z*: 500.3740.

1,2,3,4,6-Pentaacetyl α/β-l-mannopyranoside (**23**)

l-Mannose (2.00 g, 11.10 mmol), was dissolved in pyridine (40.0 mL), then acetic anhydride (11.00 mL, 111.00 mmol) was added followed by dimethyl amino pyridine (DMAP, 0.27 g 2.20 mmol). The mixture was stirred vigorously for 18 h at room temperature and it was stopped by addition of methanol (10.0 mL) and then stirred for 15 min. The solvents were removed under high vac. The resulting residue was then dissolved in ethyl acetate (50.0 mL) and washed with 3% HCl solution (1 time), saturated sodium bicarbonate (2 times), distilled water (1 time) and brine (1 time). The resulting organic layer was dried over Na_2_SO_4_ and concentrated to dryness and purified by flash chromatography using ethyl acetate and hexane (1:1) to give **23** (3.70 g, 9.48 mmol) as α, β mixture (4:1). Yield was 85%. NMR data for α- anomer of compound **23:**
^1^H-NMR (300 MHz, chloroform-*d*) δ 5.94 (d, *J* = 1.9 Hz, 1H, H-1), 5.24–5.06 (m, 3H, H-2), 4.14 (dd, *J* = 12.7, 4.9 Hz, 1H, H-6_a_), 4.05–3.85 (m, 2H, H-5, H-6_b_), 2.09 (s, 3H), 2.01 (s, 3H), 1.96 (s, 3H), 1.92 (s, 3H), 1.82 (s, 3H). ^13^C-NMR (75 MHz, CDCl_3_) δ 170.34, 169.73, 169.50, 169.34, 167.88, 90.44, 70.45, 68.63, 68.20, 65.39, 61.94, 20.62, 20.53, 20.48, 20.44, 20.41. ES-MS: calcd: C_16_H_22_O_11_Na^+^
*m*/*z*: 414.1, found [M + Na]^+^
*m*/*z*: 414.5.

Phenyl-2,3,4,6-tetra-*O*-acetyl-1-thio-α-l-mannopyranoside (**24**)

The pentaaacetate **23** (1.50 g, 3.35 mmol) was dissolved in in 30.0 mL DCM, then thiophenol (1.30 g, 11.50 mmol) and BF_3_·Et_2_O (1.60 g, 11.50 mmol) were sequentially added. The reaction was stirred vigorously for 18 h after which it was stopped with 30 mL saturated sodium bicarbonate at 0 °C. The organic layer was separated using separatory funnel and subsequently washed with saturated 35.0 mL of sodium bicarbonate (2 times), 30.0 mL of water (1 time) and 30.0 mL of brine (1 time). The organic layer was dried over anhydrous sodium bicarbonate and then concentrated in vacuo. The residue was then purified by flash chromatography using ethyl acetate and hexane (4:6) to give mainly α-anomer of **24** (0.99 g, 2.35 mmol). Yield was 70%. NMR data of compound **24** is similar to previously reported data [6]. ES-MS: calcd: C_20_H_24_O_9_Na^+^
*m*/*z*: 463.1, found [M + Na]^+^
*m*/*z*: 462.9.

Phenyl-1-thio-α-l-mannopyranoside (**25**)

Compound **24** (1.00 g, 2.35 mmol) was dissolved in 15.0 mL of methanol, then catalytic amount of sodium methoxide was added and the reaction was vigorously stirred for 3 h. The reaction was stopped by acidic ion exchange resin. The resin was filtered and the filtrate was concentrated under vacuum and the residue was dissolved in ethyl acetate and filtered again through a pad of silica gel to give **25** (0.58 g, 2.14 mmol) as a white solid. Yield was 91%. This compound was used without further purification and characterization for the next step.

Phenyl-6-tosyl-1-thio-α-l-mannopyranoside (**26**)

Compound **25** (1.31 g, 4.80 mmol) was dissolved in dissolved in 20.0 mL pyridine, cooled to 0 °C, then toluenesulphonychloride (1.06 g, 5.54 mmol) and DMAP (0.05 g, 0.41 mmol) were added. The temperature was allowed to increase to room temperature and the mixture was stirred for 18 h. At the end of reaction, the mixture was diluted with methanol after which the solvents were removed in vacuo. The residue was diluted with 40.0 mL ethyl acetate and the washed sodium bicarbonate solution (3 times), brine (3 times). The organic layer was then dried over sodium sulphate and then concentrated under vacuum and the residue was partially purified by flash chromatography using ethyl acetate (100%) to give **26** (1.81 g, 4.24 mmol) as a white solid. Yield was 88%. **26** was not characterized using NMR. ES-MS: calcd: C_19_H_22_O_7_S_2_Na^+^
*m*/*z*: 449.1, found [M + Na]^+^
*m*/*z*: 449.5.

Phenyl-6-azido-1-thio-α-l-mannopyranoside (**27**)

Compound **26** (1.81 g, 4.24 mmol) and sodium azide (1.80 g, 27.60 mmol) were suspended in anhydrous DMF and the mixture was stirred at 90 °C for 18 h. at the end of the reaction the mixture was concentrated then diluted with ethyl acetate and filtered to remove excess sodium azide. The filtrate was then concentrated and partially purified with flash chromatography using hexane/ethyl acetate (1:9) to give **27** (1.20 g, 4.04 mmol) as a white solid. Yield was 95%. ES-MS: calcd: C_12_H_15_N_3_O_4_SNa^+^
*m*/*z*: 320.1, found [M + Na]^+^
*m*/*z*: 320.3

Phenyl-2,3,4-tri-*O*-acetyl-6-azido-1-thio-α-l-mannopyranoside (**28**)

Compound **27** (1.20 g, 4.04 mmol) was dissolved in pyridine (50.0 mL), then acetic anhydride (2.00 mL, 20.00 mmol) was added followed by dimethyl amino pyridine (DMAP, 0.05 g, 0.41 mmol). The mixture was stirred vigorously for 18 h at room temperature and it was stopped by addition of methanol (10.0 mL) and then stirred for 15 min. The solvents were removed under high vac. The resulting residue was then dissolved in ethyl acetate (50.0 mL) and washed with 3% HCl solution (1 time), saturated sodium bicarbonate (2 times), distilled water (1 time) and brine (1 time). The resulting organic layer was dried over Na_2_SO_4_ and concentrated to dryness and purified by flash chromatography using ethyl acetate and hexane (1:1) to give **28** (1.31 g, 3.10 mmol) white solid. Yield was 76%. NMR data of compound **28**: ^1^H-NMR (300 MHz, chloroform-*d*) δ 7.54–7.10 (m, H, aromatic protons), 5.54–5.37 (m, 2H, H-1, H-3), 5.32–5.19 (m, 2H, H-2, H-4), 4.43–4.38 (m, 1H, H-5), 3.41–3.17 (m, 2H, H-6), 2.07 (s, 3H), 2.03 (s, 3H), 1.90 (s, 3H). ^13^C-NMR (75 MHz, CDCl_3_) δ 169.72, 169.64, 132.49, 132.01, 129.27, 128.12, 85.64, 71.01, 70.81, 69.18, 67.14, 50.98, 20.69, 20.58, 20.50. ES-MS: calcd: C_18_H_21_N_3_O_7_SNa^+^
*m*/*z*: 446.1, found [M + Na]^+^
*m*/*z*: 446.4.

3-Azido-1-hexadecyloxyl-2*R*-(6′-azido-2′3′4′-tri-*O*-acetyl-*O*-α-l-mannopyranosyl)-*sn*-glycerol (**29**)

The fully protected glycoside donor **28** (0.20 g, 0.47 mmol) and the glycoside acceptor **15** (0.18 g, 0.52 mmol) were dissolved in 15.0 mL of DCM under argon atmosphere, then AgOTf (0.02 g, 0.09 mmol) and N-iodosuccinimide (0.16g, 0.71 mmol) were simultaneously added. The reaction was vigorously stirred for 2 h after which it was stopped with saturated solution of sodium thiosulphate (5.0 mL) and then washed with 25.0 mL of saturated sodium thiosulphate solution (1 time), saturated sodium bicarbonate (3 times), water (1 time) and brine (1 time). The organic layer was then dried over anhydrous sodium sulphate and then concentrated under vac. The residue was purified by flash chromatography using ethyl acetate/hexane mixture (4:6) to give **29** (0.220 g, 0.34 mmol) as a white solid. Yield was 71%**.** NMR data of **29**: ^1^H-NMR (300 MHz, chloroform-*d*) δ 5.38 (dd, *J* = 9.9, 3.3 Hz, 1H, H-3), 5.32–5.20 (m, 2H, H-2, H-4), 5.06 (d, *J* = 1.8 Hz, 1H, H-1), 4.20 (ddd, *J* = 9.5, 5.6, 3.4 Hz, 1H, H-5), 4.07–3.89 (m, 1H), 3.64–3.23 (m, 8H, H-6), 2.18 (s, 3H), 2.06 (s, 3H), 2.01 (s, 3H), 1.61–1.54 (m, 2H), 1.27 (broad s, 26H, lipid tail), 0.89 (t, *J* = 6.5 Hz, 3H, lipid terminal-CH_3_). ^13^C-NMR (75 MHz, CDCl_3_) δ 170.01, 169.81, 97.13, 76.84, 71.80, 70.35, 70.23, 69.69, 68.66, 67.20, 51.76, 51.18, 31.93, 29.70, 29.64, 29.52, 29.37, 26.15, 22.70, 20.88, 20.72, 14.12. ES-MS: calcd: C_31_H_54_N_6_O_9_Na^+^
*m*/*z*: 677.4, found [M + Na]^+^
*m*/*z*: 677.8.

3-Azido-1-hexadecyloxyl-2*R*-(6′-azido-*O*-α-l-mannopyranosyl)-*sn*-glycerol (**30**)

Compound **29** (0.22 g, 0.34 mmol) was dissolved in 15.0 mL of methanol, then catalytic amount of sodium methoxide (0.05 g) was added and the reaction was vigorously stirred for 3 h. The reaction was stopped by acidic ion exchange resin. The resin was filtered and the filtrate was concentrated under vacuum and the residue was purified by flash chromatography using ethyl acetate/hexane mixture (9:1) to give **30** (0.120 g, 0.23 mmol) as a white solid. Yield was 68%. NMR data of **30**: ^1^H- NMR (300 MHz, methanol-*d*_4_) δ 5.02 (d, *J* = 1.9 Hz, 1H, H-1), 4.10–3.89 (m, 2H, H-3), 3.96–3.81 (m, 2H, H-2), 3.78–3.57 (m, 3H), 3.56–3.28 (m, 6H), 1.62–1.58 (m, 2H), 1.33 (broad s, 26H, lipid tail), 0.91 (t, *J* = 6.6 Hz, 3H, lipid terminal-CH_3_). ^13^C-NMR (75 MHz, MeOD) δ 101.93, 77.92, 74.29, 72.72, 72.18, 72.15, 71.65, 69.45, 53.23, 52.92, 33.16, 30.89, 30.79, 30.57, 27.33, 23.82, 14.59. ES-MS: calcd: C_25_H_48_N_6_O_6_Na^+^
*m*/*z*: 528.4, found [M + Na]^+^
*m*/*z*: 528.7.

3-Amino-1-hexadecyloxyl-2*R*-(6′-amino-6′deoxy-*O*-α-l-mannopyranosyl)-*sn*-glycerol (**6**)

To a solution of compound **30** (0.12 g, 0.23 mmol) in THF (7.0 mL) was added 1.5 mL of water and 2.6 mL of 1 M trimethylphosphine in THF. The reaction was vigorously stirred for 2 h at room temperature after which it was concentrated under vacuum. The residue was purified by C-18 column using gradient elution with water/methanol to give **6** (0.07 g, 0.14 mmol) as a white solid. Yield was 62%. NMR data for **6**: ^1^H-NMR (300 MHz, methanol-*d*_4_) δ 4.83 (d, *J* = 2.0 Hz, 1H, H-1), 3.79 (dd, *J* = 5.0, 2.0 Hz, 1H, H-2), 3.71–3.62 (m, 2H), 3.58–3.43 (m, 3H, H-5), 3.38–3.17 (m, 4H), 2.94–2.82 (m, 1H), 2.83–2.60 (m, 2H, H-6), 1.59–1.38 (m, 2H), 1.21 (broad s, 26H, lipid tail), 0.83 (t, *J* = 6.6 Hz, 3H, lipid terminal-CH_3_). ^13^C-NMR (75 MHz, MeOD) δ 101.52, 79.00, 74.36, 72.75, 72.70, 72.44, 72.34, 69.51, 43.45, 43.26, 33.12, 30.84, 30.70, 30.52, 27.37, 23.78, 14.51. MALDI-HRMS: calcd: C_25_H_52_N_2_O_6_Na^+^
*m*/*z*: 499.3723, found [M+Na]^+^
*m*/*z*: 499.3409.

### 4.2. Biological Methods

#### 4.2.1. Determination of Cytotoxicity of GAELs on Cancer Cell Lines

The cell lines were cultured from frozen stocks originally obtained from ATCC (Manassas, VA, USA). MDA-MB-231, JIMT-1, DU145, MDA-MB-468, Hs578t and MDA-MB-453 were grown in DMEM medium supplemented with 10% FBS. BT-474 cells were grown in DMEM/F12 medium supplemented with 10% FBS. MiaPaCa2 was cultured in DMEM supplemented with 10% FBS and 2.5% horse serum. All the media contained penicillin/streptomycin. U-251 and U-87 in Eagle MEM supplemented by 10% FBS, 1% Non-essential amino acids, 1 mM sodium pyruvate and 2 mM glutamine. BT-549 was cultured in RPMI 1640 medium supplemented with 10% FBS.

The effects of the GAELs on the viability of the epithelial cancer cell lines were determined as previously described [4,6,7]. Briefly, equal numbers of the cells were dispersed into 96-well plates. After 24 h, the cells were incubated with the compounds (0–30 µM) for 48 h. MTS reagent [3-(4,5-dimethylthiazol-2-yl)-5-(3-carboxymethoxyphenyl)-2-(4-sulfophenyl)-2*H*-tetrazolium, inner salt; Promega, Madison, WI, USA] (20% *v*/*v*) was subsequently added and the plates were incubated for 1–4 h in a CO_2_ incubator followed by measurement of the medium at OD_490_ with a plate reader (Molecular Devices). Wells with media but no cells were treated in similar fashion and the values utilized as blank. The results represent the mean ± standard deviation of 6 independent determinations.

#### 4.2.2. Determination of Membranolytic Effects of GAELs

Equal numbers of JIMT-1 and DU145 cells were dispersed into 96-well plates. After 24 h the cells were incubated with varying concentration of compounds **3** or **4** (4–6 µM) for 5–6 h. Subsequently, 2 µM of the cell membrane impermeant dye, ethidium homodimer-1 (EthD-1 (Molecular Probes, Eugene, OR, USA) that emits red fluorescence upon binding to DNA was added and the cells analysed by fluorescence microscopy. EthD-1 staining was compared to negative controls with no treatment and positive control treated with 0.01% Triton X-100 for 10 min.

#### 4.2.3. Demonstration of Caspase-mediated-apoptosis Independent Mode of Cell Death

Equal numbers of JIMT-1, or DU145 cells were dispersed into 96-well plates, and after 4 h the cells were treated with pan-caspase inhibitor QVD-OPh (40 µM). After 20 h, the cells were incubated with the varying concentration of the compounds (0–9 µM) for 48 h. At the end of the incubation, the MTS assay was used to assess cell viability as described above. The results represent the mean ± standard deviation of 6 independent determinations.

#### 4.2.4. Hemolytic Assay

The hemolytic activity of the GAEL analogs was evaluated using ovine erythrocyte. Sheep whole blood was collected from a slaughter house into a vessel containing disodium EDTA (1.2 g/mL) in a buffered saline (10 mM Tris, 150 mM NaCl, pH 7.4). The erythrocytes were prepared and washed with buffered saline as previously reported [4]. For the assay, the erythrocyte suspension, varying amounts of the GAEL drugs or vehicle were pipetted into 1.5 mL microcentrifuge tubes to give a final volume of 1500 µL and cell density of 2.5 × 10^8^ cells/mL. The suspensions were incubated with gentle shaking in Eppendorf thermomixer for 30 min. The 1.5 mL microcentrifuge tubes were cooled in ice water and centrifuged at 2000× *g* and 4 °C for 5 min to pellet cell debris. 200 µL of the supernatant was dissolved in 1800 µL of 0.5% NH_4_OH and the optical density (OD) was recorded using 1 mL cuvette at 540 nm in a spectrophotometer. For 0% hemolysis, buffer and vehicle were added instead of the drug, and 1% NH_4_OH was used to generate 100% hemolysis. The % hemolysis was calculated using the optical density (OD) values as shown below in Equation (1) [4]:% hemolysis = (X − 0%)/(100% − 0%)(1)
where X is the OD values of the drug(s) at varying concentration.

#### 4.2.5. Statistical Analysis

The results represent the mean ± standard deviation of 6 independent determinations. Statistical significant difference test was carried using GraphPadInstat software (GraphPad Software, San Diego, CA, USA). The mean values were subjected to one-way analysis of variance (ANOVA) followed by Tukey-Kramer multiple comparison post hoc test. Comparisons were carried out between the viability of controls and drug treated cells to determine if statistically significant differences existed between the two groups. The results of the effects of different concentrations of the compounds were also compared for statistically significant differences to determine if the cytotoxic activities of the drugs are dose dependent. The anticancer activities of all the compounds **2**–**6** tested were compared using ANOVA followed by Tukey-Kramer multiple comparison tests at the following concentrations: 5, 7.5 and 10 µM to determine if the difference in the potency of the drugs are statistically significant or not. A *p* value > 0.05 indicates no statistical differences while a *p* value <0.001 indicated statistical significant differences. The statistical analysis data are not included in this report.

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
