# Peer review of "Syntheses of l-Rhamnose-Linked Amino Glycerolipids and Their Cytotoxic Activities against Human Cancer Cells"

_molecules, 2020, doi:10.3390/molecules25030566_

Round 1

Reviewer 1 Report

I enjoyed reading this valuable investigation about syntheses of new compounds to treat cancer (in in-vivo scale) by Makanjuola et al. Just to improve the readability of the menuscript I have some minor comments.

In this sentence in Abstract complete name of the compound is missing. Introduction is written so concise. More information would improve the readability and readers would be more informed and follow the paper. The results in Figures 2,3, and four presented as a continuse numbers, while they should be discrete. It would be nice if Authors quantify the significant difference of compound activities and demonstrate it as percentage. For example,  L-rhamnose GAEL 3 is ...% more lethal than compounds X,Y, and Z. For my curiosity, I want to know how Authors picked these three cell lines, breast, pancreas, and prostate. For me, breast cancer is one of the most heterogeneous cancer, pancreas cancer is a bad prognosis cancer and prostate cancer is a good one. Is there any biological reason for picking these cell lines? Is it possible to treat the normal counterparts of these cell lines by the L-rhamnose GAEL 3 and present the result in the next version of the manuscript?

Author Response

We are unsure what the Reviewer is alluding to as the Abstract does in fact provide the complete name of the most active analog.

While we do not dispute the assertion that the presenting the data as discrete numbers is desirable, for ease of presentation, they were presented as continuous lines. Presentation of data as continuous lines is an acceptable format and is widely used in virtually all scientific journals. 

The Reviewer suggested that we quantify the differences between L-rhamnose-GAEL and other compounds as percentages in the text. In light of the number of cell lines that were tested, it would be cumbersome to describe the differences between the rhamnose-GAEL and each compounds as a percentage for each cell line. We have therefore made comparisons and summarized the differences observed by providing the range of differences (presented as a fold difference) that were observed between the rhamnose GAEL and other compounds for all cell lines

The cancer cell lines picked were selected to demonstrate the broad applicability of the GAELs. As the Reviewer rightly pointed out these cells represent a broad section of available cancer types.

Reviewer 2 Report

Well presented research article of high importance.

I would recommend to modify the graph legend of figures 2 and 3.

It is difficult to distinguish among the compounds toxicity data. 

Author Response

The Reviewer’s suggestion is welcome and has been implemented.  The symbols for Figures 2 and 3 have been colour-coded to allow one to distinguish the toxicity of the different compounds (Fig 2) and cell lines (Fig 3).

Reviewer 3 Report

Makanjuola et al described the syntheses of L-Rhamnose linked amino glycerolipids and examine their activities agains several human cancer cell lines.  The authors have shown that compound 3 is the most potent compound at inhibiting cell growth & toxicity.  The authors further investigated the potential mechanism of action of compound 3.

Overall, the syntheses and presentation of the each steps and final characterization are good.  Although, are all compounds sodium salt?  All mass spec reported showed sodium ion adduct, which is rare in LC-MS (unless the water is not HPLC grade).

The major problem of the manuscript is in the biological characterization.  MTS assay can be use to determine growth inhibition and cytotoxicity.  However, the data need to be interpreted carefully.  Unless a baseline reading is take at time zero, after 24-48 hrs, the baseline signal will be very low even if the compound simply inhibits the growth of the cell.  For example, please see the NIH 60 cell line screen protocol (https://dtp.cancer.gov/discovery_development/nci-60/methodology.htm).  MTS can be substituted for the detection method described in the methodology.  The caspase-independent assay is good, but there is no other data to support so called "methuosis" described by the author.  There are many caspase-independent cell death.  Of course, the compounds can simply be stopping cell proliferation.  If cells are died, ethidium assay in figure 5 will be positive (it by itself is a way to check cell viability under microscopy).  Thus, I do not believe the authors conclusion is correct.

Author Response

We used electrospray to detect the molecular ions of the compounds. We frequently observe sodiated molecular ions of our carbohydrate-based compounds in electrospray. This is not unusual and has been reported previously by us. In this case the sodiated molecular ion peaks were the most predominant and likely is the result that we stored in compounds in glass vials.

We do agree that the MTS assay can be used to determine both growth inhibition (proliferation) and cytotoxicity. Nevertheless, the assay is a universally accepted assay to measure cell viability.  It is worth pointing out that in our assays we had wells with no cells that were treated in an identical fashion as the experimental wells with cells.  This was clearly stated in the methodology in the manuscript. Thus, incubation conditions that lead to the absence of any viable cells could be identified.  Therefore in the graphs where viability is shown as zero, there are no viable cells and all cells are dead, having been killed by the compounds.  This demonstrates the cytotoxic effects of the GAEL.

We do agree with the referee that we did not present data to support the possibility of the mechanism of action being ‘methuosis’. We never presented this as a definite proven fact. What we wrote was that ‘ it may be similar to …’ (line 260), based on the similarity of the characteristics observed with other GAELs. 

With respect to the ethidium bromide experiments, the Reviewer is actually concurring with our view that the results in Figure 5 prove that the cells were not dead. It should be borne in mind that incubations were for 6 h only. The point we made was that in spite of the gross morphological changes that the compound had on the cells, this occurred in the absence of loss of membrane permeability.  Thus loss of membrane permeability is not the primary mode of action of the GAELs. 

Reviewer 4 Report

1 It appears to be claimed that  compound 14 is a single enantiomer with defined stereochemistry at C2 rather than a racemic mixture. There is no evidence or literature citation to support this - nor is there any specific rotation of this compound or anyother compound. There is also no evidence presented that any subsequent glycerol derivative used for the anomeric position is a single enantiomer - do the authors claim otherwise? 

2 This has the consequence that all claims of the anomeric derivatives of all the compounds are diastereomeric mixtures of the glycerol substituents rather than one compound. The claims that any of the compounds are pure single compounds appears to be without any merit or evidence. Thi would seem o be a few that renders all claim in is paper invalid. 

3 The SI has spectra that are not helpful in telling that they are one compound or a diastereomeric mixture of the epimers at C2 of the glycerol derivatives. The NMR analyses are not helpful since almost all of the H are multiplets; some of the assignments do not seem secure. From the data presented it is not clear there is a single diastereogmer - actually the contrary view is most likely. The analysis of the NMR spectra needs to be much more detailed; is there any evidence for which of the rhamanose anomers is formed

4 On the basis of the amount of the glycerol derivatives used, it would seem that unless the authors are sure of the enantiomeric purity, they on the basis of yield are bound to be diastereomeric mixtures

5 In lines 279-280 ..."Purity of compound 2-6 was assessed by elemental analysis of elements (C, H N) and were within ± 0.5 % of the theoretical values."  In neither the experimental nor the SI for these compounds, no data is presented - 0.5%is well outside of the conveiona requirements. There are NO cases in which any elemental analyses are presented in the experimental section or the SI

5 The nomenclature is completely inaccurate and not acceptable at any level. While there are lots of trivial mistakes there are also many fundamental flaws which make the names nonsense.

6 There are many other issues that need to be raised but the fundamental flaw that appears to be present means that this present data is not publishable in any journal

Author Response

1) Compound 14 was prepared from compound 13 which was commercially obtained as a single enantiomer. Therefore all reported compounds exist as a single stereoisomer

2) All final compounds were obtained in pure form as a single diastereoisomer. 

3 and 4) As pointed out above there is no evidence for diastereomeric mixtures as we started the synthesis from enantiomeric pure starting materials. Moreover, our C13 NMR do not indicate any evidence for the presence of diastereomeric compounds.  

5) Purity was assessed by 1H-NMR spectra as described in the instructions of the journal. We removed the sentence that purity was assessed by elemental analysis in the manuscript which was incorrect in the previous version. 

5) We have performed a nomenclature check on the compounds.

6) We disagree together with the other three reviewers of the paper. 

Round 2

Reviewer 3 Report

None.

Author Response

we have performed a spell check on the manuscript 

Reviewer 4 Report

The authors state they have purchased a single enantiomer 13 but have neither stated its source nor any literature data associated with the specific rotation. In the preparation of the tosylate 14 on lines  354 to 356...The lipid diol 13 (2.00 g, 6.32 mmol) was dissolved in dissolved [sic!] in 20 ml DCM, cooled to 0ï‚°C, then Et3N (1.28 g, 1.80 ml,) was added followed by 4-toluenesulphonyl chloride (1.33 g, 6.95 mmol) and DMAP (0.04 g, 0.32 mmol)..., it would be difficult for the authors to argue these are not conditions where base catalysed ester migration might lead to partial racemisation. There is no specific rotation given of either the tosylate 14 nor of the azide 15. Would they mind if it was zero? Probably not but why not report it? If they are new compounds, they need to state the data; if they are not new compounds, they need to give the reference where it was reported previously. 

But the key point is whether they can claim optical purity or allow anyone who repeats their work to ensure they have the same compound; they have not reported specific rotations for these new[?] compounds, it would seem to be a required piece of evidence they must give in a chemical journal.

There are still errors in the nomenclature.

Author Response

We provided the reference (reference 6) for the synthesis of known azide 15 in the manuscript. As indicated previously there was no indication for partial racemization of the 2-position of the glycerol moiety.

Selective tosylation of a primary hydroxyl group in the presence of a secondary hydroxyl group are well known. We have never observed tosylate migrations of this position under these conditions. If tosylation wound occurs the migrated product (regiosimer) can easily be distinguished in the 1H-NMR.    

Glycosylation of azide 15 with L-rhamnose-based glycosyl donor 10 as well as many other donors provides a single disastereomer together with some hydrolysed donor. Again this is a strong indication that no racemization occurred during tosylation and tosyl substitution. 

We have corrected the nomenclature and performed a spell check.